# Demography-Oriented Urban Spatial Matching of Service Facilities: Case Study of Changchun, China

**Yingzi Chen [1], Yaqi Hu [2,*] and Lina Lai [3]**

1   Northeast Asian Research Center, Jilin University, Changchun 130012, China
2   Northeast Asian College, Jilin University, Changchun 130012, China
3   Changchun Municipal Engineering and Research Institute Co., Ltd., Changchun 130022, China
*   Correspondence: yqhu20@mails.jlu.edu.cn

**Abstract:** People-oriented urban planning requires that service facilities should efficiently meet individual and community activity needs across the demographic landscape that defines a city. To develop a conceptual basis for urban spatial infrastructure optimization, we empirically studied existing population activities and service facilities in Changchun, China, using kernel density estimation, bivariate spatial autocorrelation analysis, and other models. The spatial relationships we derived from multiple sources of big data such as mobile phone signaling and POI data indicated that the intensity of population activity has obvious temporal regularity, and its spatial distribution is "center-periphery." Service facilities display a "One main and two subs" distribution with no obvious spatial dependence between the core's density and diversity. Population activities and service facility diversity show a high-high spatial correlation and multiple matching patterns. At the same time, a certain degree of spatial mismatch between different age groups and service facilities was also observed. Our research suggests several urban renewal actions to rectify this mismatch, such as: decentralizing the core area medical service facilities; reducing the attractiveness of the core area and its traffic pressure; and renewing and renovating old facilities to reduce construction costs. At a government planning level, construction along the periphery of the urban can enrich the diversity of its service facilities to improve the efficiency of spatial allocation.

**Keywords:** multi-source big data; urban population activities; service facilities; spatial analysis; pattern matching



## 1. Introduction

China's urban spatial structure planning is gradually changing from growth-driven outward expansion to current concerns about optimizing the internal spatial structure of cities [1]. For example, the Ministry of Natural Resources issued the 2020 "Guidelines for the Preparation of Municipal Territorial Spatial Master Plan (for Trial Implementation)", which emphasizes the need to improve public space and public service functions and to enhance the quality of the living environment [2]. Urban population activities are mainly divided into production and living activities [3], and service facilities are essential for the production and living activities of urban population [4]. The spatial layout of the city is shaped by the number and distribution of these facilities and their dynamic relationship to the demography and number of urban residents [5,6], and urban residents' activities [7,8]. The greater the variety of service facilities in a region, the more active its population. The spatial layout of service facilities affects the speed and quality of the city's economic development. The daily interaction between residents and service facilities determines the spatial vitality of cities [9,10], and the coordinated matching of urban services and residents' living spaces can effectively alleviate urban problems such as air pollution and traffic congestion [11,12]. In 2022, the National Development and Reform Commission issued the "2022 New Urbanization and Urban-Rural Integration Development Key Tasks",

which proposed to accelerate the construction of smart cities [13]. Smart cities require the layout of service facilities to be more responsive to the needs of the population [14–16]. By collecting big data such as mobile phone signaling and POI, smart cities can analyze regional differentiated demand and the supply capacity of regional facilities, and promote the optimization of the spatial layout of the city.

The spatial layout of cities is closely related to population activities [17]. The scale of population movement between cities in China is increasing, and pressure on urban service facilities is now coming from residents and short-stay tourists [18,19]. According to the Seventh National Population Census, China's floating population was 376 million in 2020, and their living needs should be met within cities. China's tourism industry has been hit by COVID-19 in recent years, according to data breakdowns. Using 2019 data, we found that China had 6006 million domestic tourists, putting pressure on service facilities in tourist destinations [20]. Short-stay commuting and tourism populations are often neglected in urban spatial planning [21], leading to a loss of efficiency in urban spatial allocation. To accommodate these groups in the analyses, the definition of the existing population was adapted from "Certain Provisions on the Service and Management of the Existing Population in Shanghai" by relaxing the temporal restriction. The existing population is defined as the population with household registration in the region and the population with household registration outside the region who live and stay in the study area [22]. Such a definition will facilitate a more comprehensive, inclusive, and efficient rational layout of the spatial structure of the city.

Different demographic characteristics such as age [23], personal preference [24,25], race [26], and income [27] will represent different needs for urban service facilities. Based on the traditional locational allocation model, the spatial planning of government service facilities only focuses on the total demand of the regional population, and the number of service facilities owned by 1000 people is used to measure the degree of matching service facilities with the population [28]. This approach ignores the individual differences of the population, resulting in a spatial mismatch between service facilities and population activities [21]. The goal of demographic-oriented spatial allocation of service facilities is to meet the diverse needs of various groups of people [29]. People of different age structures have different service needs, and most differences in population activity intensity are based on age. The age structure of the group is one of the most important factors to be considered in the spatial layout of service facilities [30]. The demographic-oriented spatial layout of service facilities is an important way to improve the spatial efficiency of the city.

The spatial distribution of service facilities should be balanced, and the scale should be appropriate to the needs of the population. The number and diversity of service facilities are important influencing factors for the choice of population activities [31]. A good interaction between population and service facilities is the basis of urban economic development [32]. Evaluating the spatial match between population and service facilities is an important basis for improving the spatial configuration of cities. Spatial matching degree refers to the degree of spatial coordination between two or more study objects. Researchers have tried to evaluate the degree of spatial matching between population activities and service facilities from various perspectives. Based on multi-source big data, scholars have found that in the spatial dimension there is a mismatch between the population and number of service facilities in large cities, and the spatial distribution is uneven [33]. In the time dimension, people's demand for service facilities on holidays is significantly different from that on weekdays [34], and the degree of spatial matching between service facilities and population is also different [35]. Season also has a significant effect on the matching of the two [36]. The above study summarized the basic laws of spatial matching of population and service facilities but did not consider the effect of age on the demand for service facilities.

"Local-scale data and urban models are required to examine the fine-scale spatial match of urban population activities and service facility distribution." Most traditional statistics of population distribution use survey data; the sample size of the survey is small, data are difficult to collect, and the statistical scale is broad [37]. Survey data are not

only unable to reflect the real characteristics of urban population activities, but also fail to identify the large number of short-stay people in the city. The use of mobile phone signaling data can remedy this deficiency [38,39]. These data can capture the spatial and temporal movement and distribution information of residents [40].

The following are contributions of this paper: First, we introduce the existing population into the scope of urban population studies, which complements the traditional statistics of the urban population and more reasonably evaluates the pressure faced by service facilities. Second, we analyze the spatial matching of service facilities from the demographic point of view, considering the different needs of different age groups, and provide a case study for optimizing the spatial allocation of cities with a human-centered approach. Finally, the types of spatial matching are subdivided to identify special spatial matching and provide a theoretical basis for refined urban planning.

## 2. Material and Methods

In this section, we present the data and methods used to assess the relationship between population activity and service supply. Our research methods included the kernel density method, Shannon–Wiener Index, and bivariate spatial autocorrelation analysis. We used the kernel density method to analyze the spatial distribution of service facilities because, compared with other methods (e.g., Quadrat analysis [41], Voronoi-based method [42]), it is based on the first law of geography, which considers locational effects and can measure the concentration of service facilities in spatial distribution [43,44]. We use the Shannon–Wiener Index based on the information–theoretic approach, which is a good measure of the diversity of service facilities [45,46]. Our spatial matching research methods included the two-step floating catchment area method [47,48], gravity model method [49,50], coupled coordination model [51], and bivariate spatial autocorrelation analysis [52] to analyze the degree of spatial matching between service facilities and population activities. Compared with other methods, the advantage of spatial autocorrelation is that it is not easily affected by the deviation of data from normal distribution and has wider applicability [53]. Therefore, we used the bivariate spatial autocorrelation analysis model to reveal the potential spatial dependence of urban population and service facilities.

As shown in Figure 1, we constructed a research framework for matching population activities with service facilities in cities. Our research included the following steps: first, we analyzed the activity characteristics of the urban population and compared the activity characteristics of different age structure groups to the spatial activity characteristics of the existing population. Second, we analyzed the spatial characteristics of service facilities in Changchun in terms of the number of regional service facilities and the diversity of service facilities. Finally, we used a bivariate spatial autocorrelation model to construct a spatial match between real population activities and service facility diversity and analyze the characteristics of different types of spatial matches. Groups were also classified according to age and used to examine the spatial dependence of different groups on different types of service facilities.

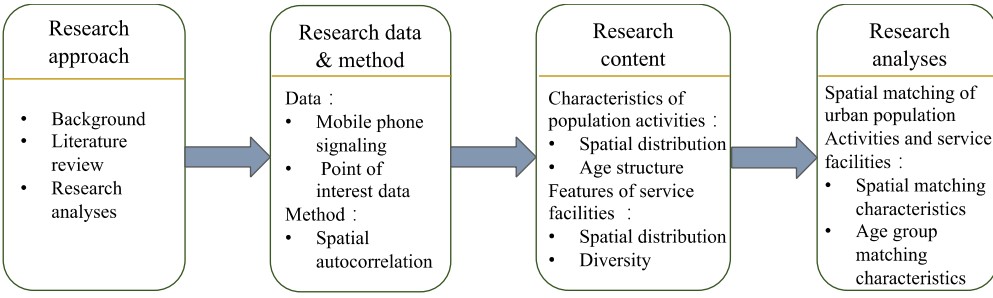

**Figure 1.** Research framework for spatial matching of service facilities and population activities.

### 2.1. Study Area

Jilin Province is an important province in Northeast China, and Changchun is the capital of Jilin Province. The main urban area is the center of population and economic concentration in Changchun. In this paper, the main urban area of Changchun was selected as defined in the Changchun Urban Master Plan (2011–2020) (the Master Plan), containing 65 districts of streets/towns (see Figure 2) [54]. According to the Seventh National Census, the total population of Changchun in 2020 was 9,066,906, with an urbanization rate of 65.94%, and the population of the main urban area accounted for about 47.8% of the total population of Changchun [55].

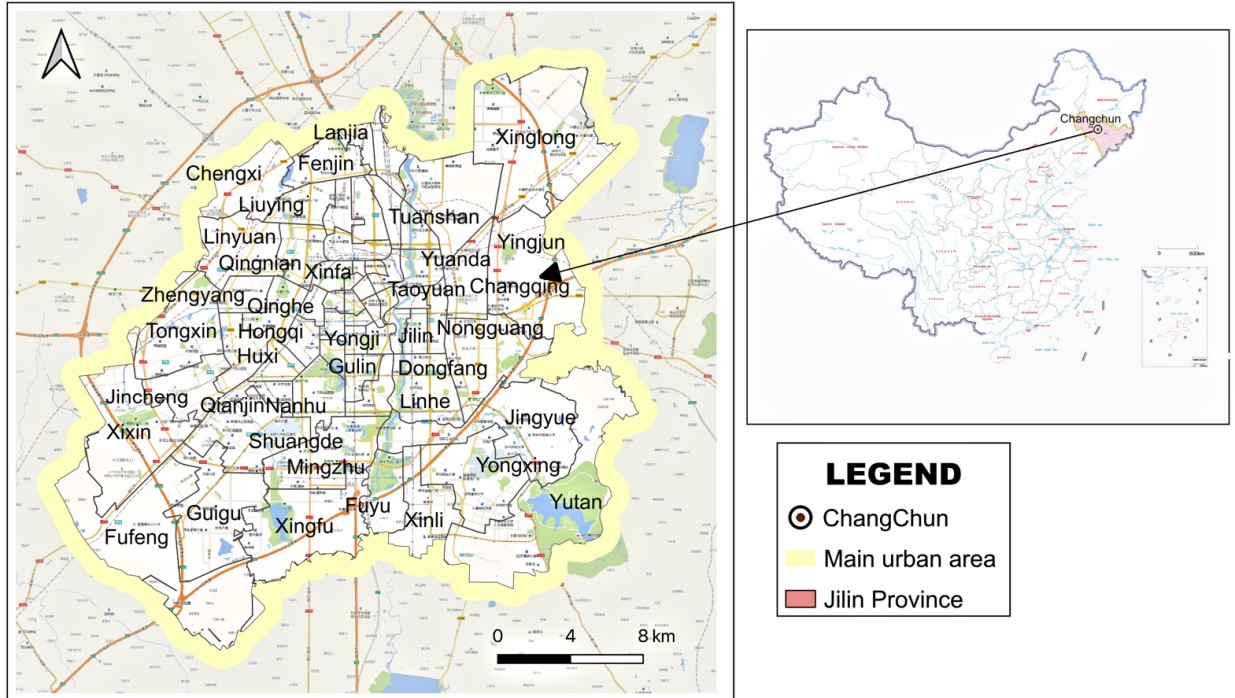

**Figure 2.** Study area. Source: Ministry of Natural Resources of People's Republic of China [56].

### 2.2. Data

We used multiple sources of big data, including not only mobile phone signaling data for describing urban population activities and identifying the existing population, but also point of interest (POI) data for describing the spatial layout and diversity of urban services.

#### 2.2.1. Mobile Phone Signaling Data

With technological progress, mobile phone ownership has increased, and in 2020, the cell phone ownership rate in China reached 1.139 phones per person [57]. In April 2021, the number of cell phones in Jilin Province reached 28.77 million. At the same time, the quantity and quality of data obtained from mobile phone signaling have improved, and mobile phone signaling data have become important in studying the spatial activities of the population. They are divided into phone list data, PS (packet switch) domain data, and CS (circuit switched) domain signaling data. We chose CS data, which contain spatiotemporal data of base station switching and location update, offering high coverage and accurately describing individual activity trajectories. China Unicom's desensitized mobile phone signaling data were selected, and data for under 18 and over 75 years old were removed owing to their poor reliability. Duplicate data and ping-pong switching data were removed, and the data missing important information values were deleted, resulting in the identification of a total of 2.77 million people, about 68% of Changchun's resident population, which was representative. There have been studies on the uniform expansion of cell phone signaling or expansion using the proportion of the existing resident

population, which has no effect on the variability of the spatial matching of the existing population and service facilities before and after expansion, so we directly used the data in the study.

### 2.2.2. Point of Interest (POI) Data

Point of Interest (POI) data refer to point data in electronic maps containing attributes such as name, coordinates, and category. In this paper, we used Python to crawl Geoda POI data in the central city of Changchun and divided it into six categories (school, restaurant, store, activity center, hospital, entertainment) according to code (see Table 1). The larger scale grid was selected because, the smaller the grid range, the fewer the number of POIs in the space and the worse the representativeness of POI diversity. At the same time, the 15 min walking distance of the population was about 1000 m. Therefore, a 1000 m × 1000 m grid was selected to calculate the POI sub-region. The data were cleaned to remove duplicate points, with 96,590 data retained and 157 grids without POIs. We removed the grids with nonexistent POIs and population activity and kept a total of 616 grids.

**Table 1.** Summary of Point of Interest Categories.

| Category | Type | Number | Proportion |
|---|---|---|---|
| School | Primary school, secondary school, university, vocational school, adult school, etc. | 275 | 0.28% |
| Restaurant | Chinese restaurants, canteen, teashop, wine shops, cafes, etc. | 31,373 | 32.48% |
| Store | Department store, grocery, clothing store, day-and-night shop, market, mall, etc. | 42,267 | 43.76% |
| Activity Center | Community cultural centers, youth centers, large cultural facilities | 10,213 | 10.57% |
| Hospital | General hospitals, specialist hospitals, community health centers, pharmacies, etc. | 9506 | 9.84% |
| Entertainment | Parks, cinemas, gymnasiums, fitness centers, etc. | 2956 | 3.06% |
| Total | | 96,590 | 100% |

Source: Gaode AMAP Inside [58].

### 2.3. Methodologies

#### 2.3.1. Population Activity Intensity

Resident travel is often a spatial and temporal activity with a certain purpose, and the agglomeration of urban residents' travel behavior can reflect their demand for service facilities. The probability of human interaction activities increases with the increase in population visits in a certain spatial area, so population visits are often used to characterize the intensity of urban population activities [59]. The intensity of urban population activity is characterized by the presence of existing population on an hourly grid. This gives an overall activity curve of:

$$\vec{M} = \left\{ \sum_{i=1}^{n} D_{i\_0}, \sum_{i=1}^{n} D_{i\_1} \ldots \ldots \sum_{i=1}^{n} D_{i\_23} \right\} \tag{1}$$

where $\vec{M}$ is the overall activity curve row vector, $\sum_{i=1}^{n} D_{i\_t}$ is the activity intensity degree in the main city of Changchun at time $t$, and $D_{i\_t}$ is the average total number of visits in $i$th grid at time $t$ ($t \in [0, 23]$).

$$D_i = \sum_{t=0}^{23} D_{i\_t} \tag{2}$$

where $D_i$ is the total one-day population activity of $i$th grid.

Population activity intensity is characterized by a clear diurnal cyclic temporal variation, and we used diurnal relative deviation to express diurnal differences, calculated as follows:

$$RD_i = \frac{2\left(\sum_{t=9}^{14} D_{i\_t} - \sum_{t=0}^{5} D_{i\_t}\right)}{\sum_{t=9}^{14} D_{i\_t} + \sum_{t=0}^{5} D_{i\_t}} \tag{3}$$

where $RD_i$ is the diurnal relative deviation of $i$th grid. The value range is $RD_i \in [0, 2]$. When $RD_i = 1$, the diurnal population activity intensity is absolutely balanced. We considered $RD_i \in [0, 0.7]$ as the low diurnal relative deviation, i.e., the intensity of population activity is higher at night than during the day; $RD_i \in (0.7, 1.3]$. as the diurnal equilibrium zone, and $RD_i \in (1.3, 2]$. was considered as high diurnal relative deviation, i.e., the intensity of population activity is higher during the day than at night. The intensity of population activity was selected for daytime hours from 9:00 to 14:00 and for nighttime hours from 0:00 to 5:00. When both daytime and nighttime population activity intensities in the grid were 0, it was meaningless to explore the diurnal relative deviation, and we excluded the grid. There were four grids in the main urban area of Changchun without diurnal population activity, accounting for 0.65% of the total grids, which had no impact on the overall analysis.

### 2.3.2. Kernel Density Method

The kernel density method is a nonparametric estimation method used to estimate the overall probability distribution density without assuming the overall distribution function; it has strong robustness and is mainly used to characterize the spatial distribution of elements. In this paper, kernel density was used to analyze the spatial distribution characteristics of service facilities.

$$f_h(x) = \frac{1}{nh} \sum_{i=1}^{n} K\left(\frac{x - x_i}{h}\right) \tag{4}$$

where $f_h(x)$ is the kernel density, and the larger the value of $f_h(x)$, the higher the density of population activity at that location. n represents the total number of grids in the study area, $K(*)$ is the kernel function, $x_i$ is the independently distributed observation, and $x$ is the mean value of the observation. $h$ is the width of the window, and the choice of the window width affects the density function. In this paper, based on the study of Wang et al. [60], the window width was chosen as 1 km.

### 2.3.3. Service Facility Diversity Index

The service facilities index, based on the Shannon–Wiener Index, is a measure of the type of service facilities available to meet the needs of residents in a region. This paper used the service facility diversity index as a measure of the structural characteristics of service facilities. The formula is as follows:

$$SW_i = - \sum_{u=1}^{6} P_{iu} ln P_{iu} \tag{5}$$

where $SW_i$ is the service facility diversity index for $i$th grid, and $P_{iu} = \frac{N_{iu}}{N_i}$ is the proportion of the $u$th type of POI to the total type of activity in $i$th grid. When $N_i = 0$, there is no POI in $i$th grid, the service facility diversity index is meaningless, thus $SW_i = 0$. When $N_{iu} = 0$ and $N_i > 0$, scholars mostly add a minimal value for $ln P_{iu}$ in their studies to avoid $ln P_{iu}$ being inoperable [61,62]. The larger the value of $SW_i$, the greater the variety of services available to the people traveling in the region and the more it can meet people's needs. The smaller the value of $SW_i$, the more homogeneous and specialized the regional service facilities are.

### 2.3.4. Spatial Match Index

Spatial autocorrelation is used to analyze the extent to which indicators are spatially correlated and differ [63], and it is divided into global spatial autocorrelation and local spatial autocorrelation [64]. As the study considered the spatial association and dependence of population activities and service facilities in Changchun, the Anseline (1995)

Bivariate Global Moran's I was chosen to show the spatial dependence of both using the following equation [65].

$$I = \frac{\sum_{i=1}^{n} \sum_{j=1}^{n} w_{ij}(x_i - \overline{x})(y_j - \overline{y})}{S^2 \sum_{i=1}^{n} \sum_{j=1}^{n} w_{ij}} \tag{6}$$

where I is the spatial autocorrelation index of $x$ and $y$, $w_{ij}$ is the spatial weight matrix, $x_i$ and $y_j$ are the observations in $i$th grid and $j$th grid, respectively, $\overline{x}$ and $\overline{y}$ are the means of the observations, $S^2$ denotes the total sample variance, n represents the total number of grids in the study area, $w_{ij}$ is a $n \times n$ matrix, when $i$th grid is adjacent to $j$th grid, $w_{ij} = 1$, otherwise $w_{ij} = 0$. Moran's I $\in [-1, 1]$. When Moran's I $> 0$ and the test is significant, the two variables are spatially matched; when Moran's I $< 0$ and the test is significant, the two variables are spatially mismatched; when Moran's I $= 0$, it means that the spatial distribution of the two variables is random, there is no correlation, and the two variables develop alone.

Bivariate global spatial autocorrelation can only reflect the average degree of aggregation of two variables in the study area and determine whether there is spatial correlation between the two variables. The bivariate local spatial autocorrelation can be used to determine the "positive spatial correlation" of "high-high" and "low-low" clusters, and the "negative spatial correlation" of "high-low" and "low-high" clusters. The spatial distribution of "negative spatial correlation" between "high-high" and "low-high" clusters can be analyzed, so in order to explore the local spatial correlation of bivariate variables, this paper adopted local spatial autocorrelation with the following formula:

$$I_i = z_i^x \sum_{j=1}^{n} w_{ij} z_j^y \tag{7}$$

where $I_i$ is the grid $i$th bivariate local Moran's I, $z_i^x = \frac{x_i - \overline{x}}{\sigma_x}$ and $z_j^y = \frac{y_j - \overline{y}}{\sigma_y}$ are the $x_i$ and $y_j$ standardized values. $\sigma_x$ and $\sigma_y$ the variances of the attribute $x$ and $y$. Based on $I_i$, a LISA (Local Indications of Spatial Association) spatial distribution map can be formed, which can discern the spatial differences in bivariate agglomerations.

## 3. Study Results

### 3.1. Spatial Characteristics of Population Activities

The intensity of population activity is one of the most important indicators of population activity characteristics, and the number of population visits measures it. According to the Clark model [66], the central area of the city is the population concentration area, and the intensity of population activity tends to decay exponentially with increasing distance.

3.1.1. Characteristics of the Temporal Distribution of Existing Population Activity

In the time dimension, the activity intensity showed a bimodal symmetric curve (see Figure 3). The curve was divided into five stages. The first phase was the silent phase: from 1:00 to 4:00 was the trough stabilization period, which was the sleeping period of Changchun residents, and the population activity intensity was between 280,000 and 250,000 people. The second stage was the awakening stage, from 5:00 to 7:00, when the population activity intensity suddenly increased from 450,000 to about 1.5 million and then fell back from 7:00 to 8:00, but maintained a high population activity intensity. The third stage was the stable stage, from 8:00 to 14:00, when the population activity intensity was stable at 1.1–1.2 million. The fourth phase was the active phase, with the evening peak hours from 15:00 to 18:00 and a peak of 1.2–1.6 million trips. The fifth stage was the weakening stage, from 18:00 to 0:00 of the next day, when the intensity of population activity dropped sharply, and the population activity intensity fell from 1 million to 600,000 person-times, with a drop rate of 160,000 person-times/h.

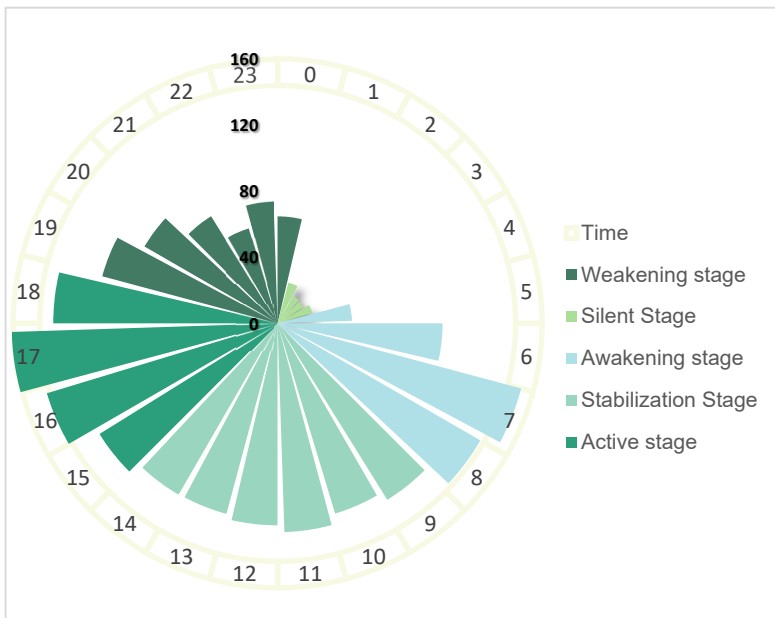

**Figure 3.** Intensity of total population activity. Note: The sectors indicate the intensity of population activity at different periods and the rings are the corresponding time points 0–23.

The activity characteristics of the different age groups are reported in Figure 4. Above the 25-year-old age group, the activity intensity of the urban population decreased with age. The activity intensity curves of the 25–55 year age group in Changchun were similar, showing a bimodal symmetry curve. Among them, the activity intensity of the 35–45 year age group was the largest, with the peak occurring at 17:00, and the population activity intensity was 498,200. The 25–35 year age group had the second highest activity intensity, and the 18–25 year, 65–75 year age groups has similar population activity intensity, and similar activity intensity curves. The activity intensity of the older working population (55–65 years old) was lower than that of the 25–55 years old activity group, but since a portion of the 55–65 years old group still had a regular job, this portion of the population had travel patterns similar to the working population in the 25–55 year age group. In addition, the non-regular working group may have provided intergenerational care for their children and transported their grandchildren to and from school. School drop-off and pick-up times for elementary and middle school students in Changchun almost coincide with the morning and evening peak periods. Population activity for the 65–75 year age group was relatively steady, with a peak of 72,200 trips at 9:00.

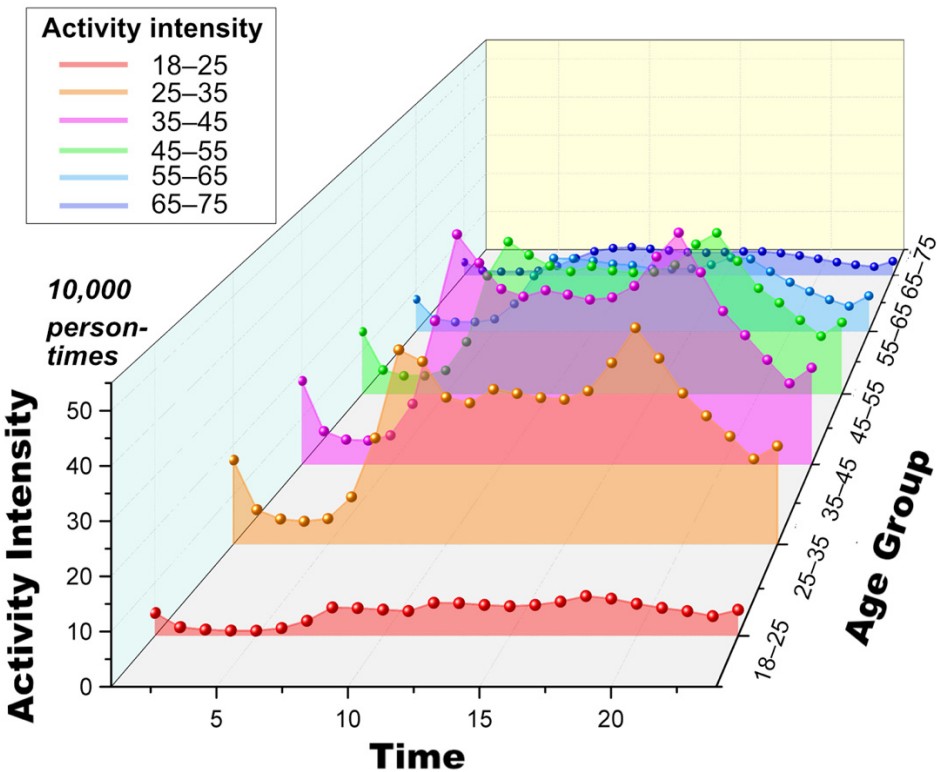

**Figure 4.** Activity intensity by age group in Changchun.

3.1.2. Characteristics of the Spatial Distribution of Existing Population Activities

As shown in Figure 5, the spatial distribution of population activities in the main urban area of Changchun was uneven. The intensity of population activity was distributed in a "center-periphery" pattern, with a circle diffusion pattern. The central area of Changchun city includes three business areas (within the blue circle), namely, Chongqing Road, Guilin Street, and Hongqi Street, and three commercial districts, namely, People's Square (a), Jilin Financial Center (c), and the Anhua Building (d). The population of Changchun was mainly concentrated within the Third Ring Area of Changchun, and the population activity within the third ring area accounted for 72.18% of the total population activity, while the population activity intensity in the urban fringe area only accounted for 27.82% of the total population activity. The Southern New City (h), as a key area for spatial optimization in Changchun, emerged as a high-density area for population activity, with POIs mostly dominated by high-end service facilities such as finance. The center of population activity in Changchun was skewed to the west, and the population activity intensity in the west was higher than that in the east. This was because the western part of Changchun's main urban area is an important industrial corridor in Changchun with numerous jobs, while the eastern part is a composite ecological corridor in the suburbs of Changchun, undertaking ecological service functions. The central area had obvious characteristics of population activity dependence and was clustered in areas with convenient transportation. By subdividing the grid, the People's Square and surrounding area (a) and the Jiemin Community (b) were the most densely populated areas in Changchun, with population activity intensities of 297,000 and 282,395, respectively. Jilin Financial Center (c) and the Anhua Building (d) followed them. The top six grids (a–f) of population activity intensity in the main urban area all exceeded 200,000 person-times. The periphery of the main urban area had lower population activity intensity, with population activity intensity below 8000 visits. Except for the area where there was no population activity, Jingyuetan National Park (g) had the lowest population activity intensity. This area is not only an important water source protection area in Changchun, but also a prohibited development area, so the population activity intensity is extremely low.

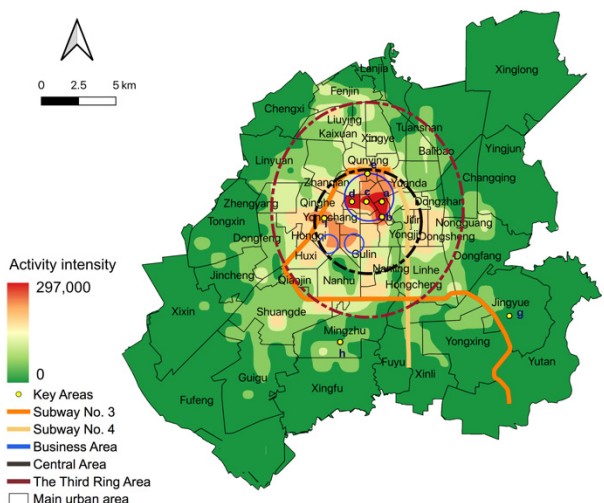

**Figure 5.** Spatial distribution of existing population activity intensity.

### 3.1.3. Spatial Characteristics of Population Activities Based on Age Groups

In terms of age groups, the distribution of population activity in the main urban area of Changchun differed among age groups. The 35–45 year age group had the highest population activity intensity (see Figure 6b), accounting for 30.08% of the total, followed by the 25–35 year age group, accounting for 25.54% of the total population activity intensity. The 18–25 and 65–75 age groups had lower population activity intensity (see Figure 6a,d), accounting for 4.72% and 4.54% of the total population activity intensity, respectively. The highest population activity intensity was found in and around the Qianwei Campus of Jilin University, with an activity intensity of 15,810 trips in the grid. In addition to this, the area around each university town was also an important concentration center. The population distribution of the 35–75 year age group had similar characteristics to the overall population, with two centers of concentration. The 35–45 year age group had the highest population activity intensity of 86,300, which was much higher than other groups. The 55–65 year age population distribution had high intensity than the 65–75 year age group (see Figure 6c,d). Many parts of the district overlapped, and the overlapping area will place more pressure on the elderly in the future, requiring reorganized layout of elderly service facilities.

As seen in Figure 7, the central and peripheral areas of Changchun's main urban area were in an obvious diurnal imbalance, dominated by high diurnal relative deviation, which meant that the intensity of population activities was higher during the day than at night. The areas with high diurnal relative deviation were distributed around commercial centers and industrial parks, which are areas with more jobs, and the population flowed into the main urban area from outside the area during the daytime, resulting in a high proportion of diurnal imbalance in the main urban area as a whole. Some areas with low relative diurnal deviation were more scattered, mostly in the areas outside the Third Ring Road, which had lower activity intensity and were mostly residential areas.

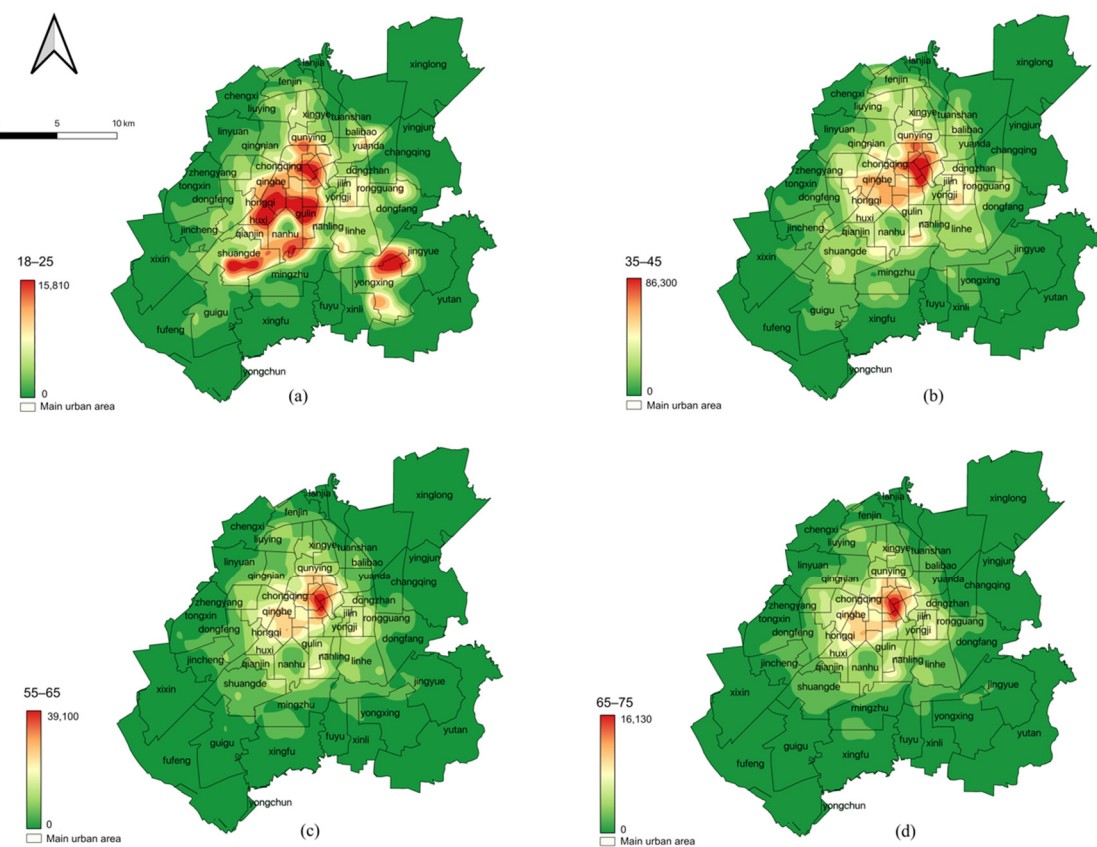

**Figure 6.** Spatial distribution of population activity intensity by age group. (**a**) 18–25; (**b**) 35–45; (**c**) 55–65; (**d**) 65–75.

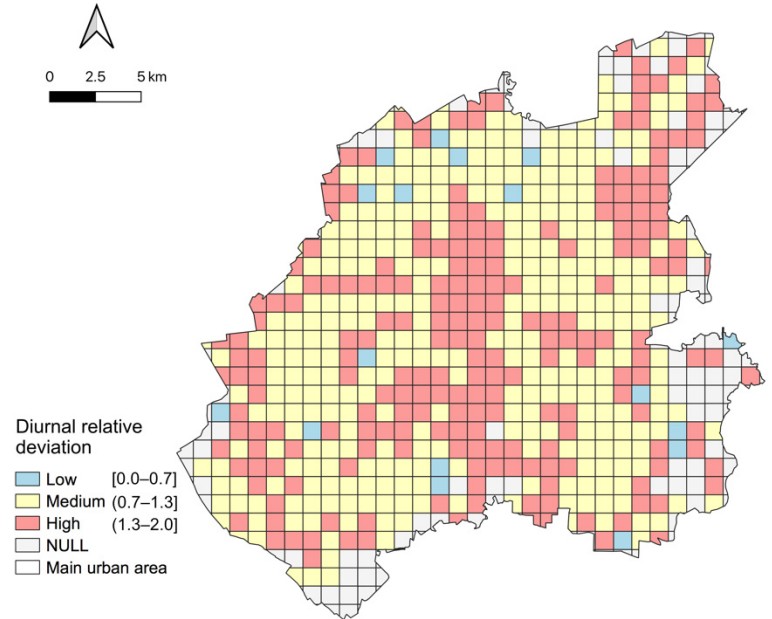

**Figure 7.** Spatial distribution of diurnal relative deviation.

### 3.2. Spatial Distribution Characteristics of Service Facilities

Population activities depend on the richness of the city's physical environment. The richer and more diversified the physical space of the city, the more types of activities there are available to the residents, the more convenient the residents' access to services, and the higher their life satisfaction. We used kernel density to analyze the spatial dis-

tribution characteristics of service facilities and color bands to characterize the change of kernel density.

### 3.2.1. Spatial Distribution of the Number of Service Facilities

From a general point of view, the service facilities in the main urban area of Changchun are distributed as "one main and two subs". As shown in Figure 8, the main center was the Chongqing Road business area with Chongqing Street, Changtong Street, and Zhanqian Street with the highest kernel density of 1970, while the sub-center was the Guilin Road business area and Hongqi Street business area, both with a kernel density of over 1000. The spatial distribution of service facilities in Changchun was intensive; Subway No. 3 and Subway No. 4 in Changchun have existed for a long time and have a profound influence on the selection of service facilities, and these service facilities formed a certain clustering trend along subway. The kernel density of service facilities decreased in a hierarchy, and the kernel density in the peripheral areas of the main urban area was much lower than that in the central area, with obvious polarization and the lowest distribution density in the southwest side. The service facilities distribution in the eastern region was lower than that in the western region, and the eastern region of Changchun mainly included ecological parks and ecological service facilities.

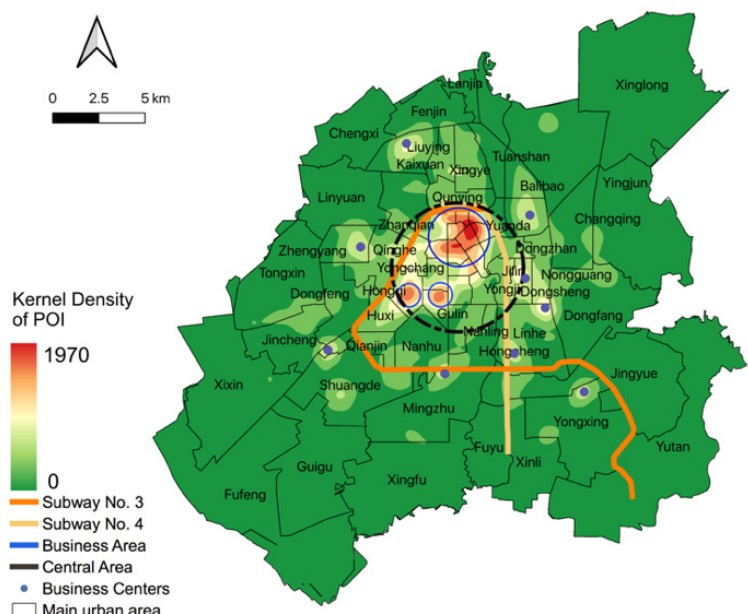

**Figure 8.** Kernel density of service facilities.

There are differences in the maximum values of kernel density for the various service facilities, which represent differences in the number of facilities for the various services. Store had the highest number of facilities with 42,267, and school had the lowest number of facilities with 275. As shown in Figure 9a–f, there were differences in the spatial distribution of different types of service facilities. Among them, the spatial distribution characteristics of food and beverage, entertainment, and overall service facility POIs remained consistent, with a high value clustering at "one main and two subs". The POIs of schools were randomly distributed and relatively evenly distributed in all areas of Changchun's main city, and the educational resources of each grid (excluding factors such as scale and grade) were relatively balanced. There was a lack of school facilities in the southwest and east of the main urban area, because the southwest is an industrial area and the east is near an ecological corridor with low population density.

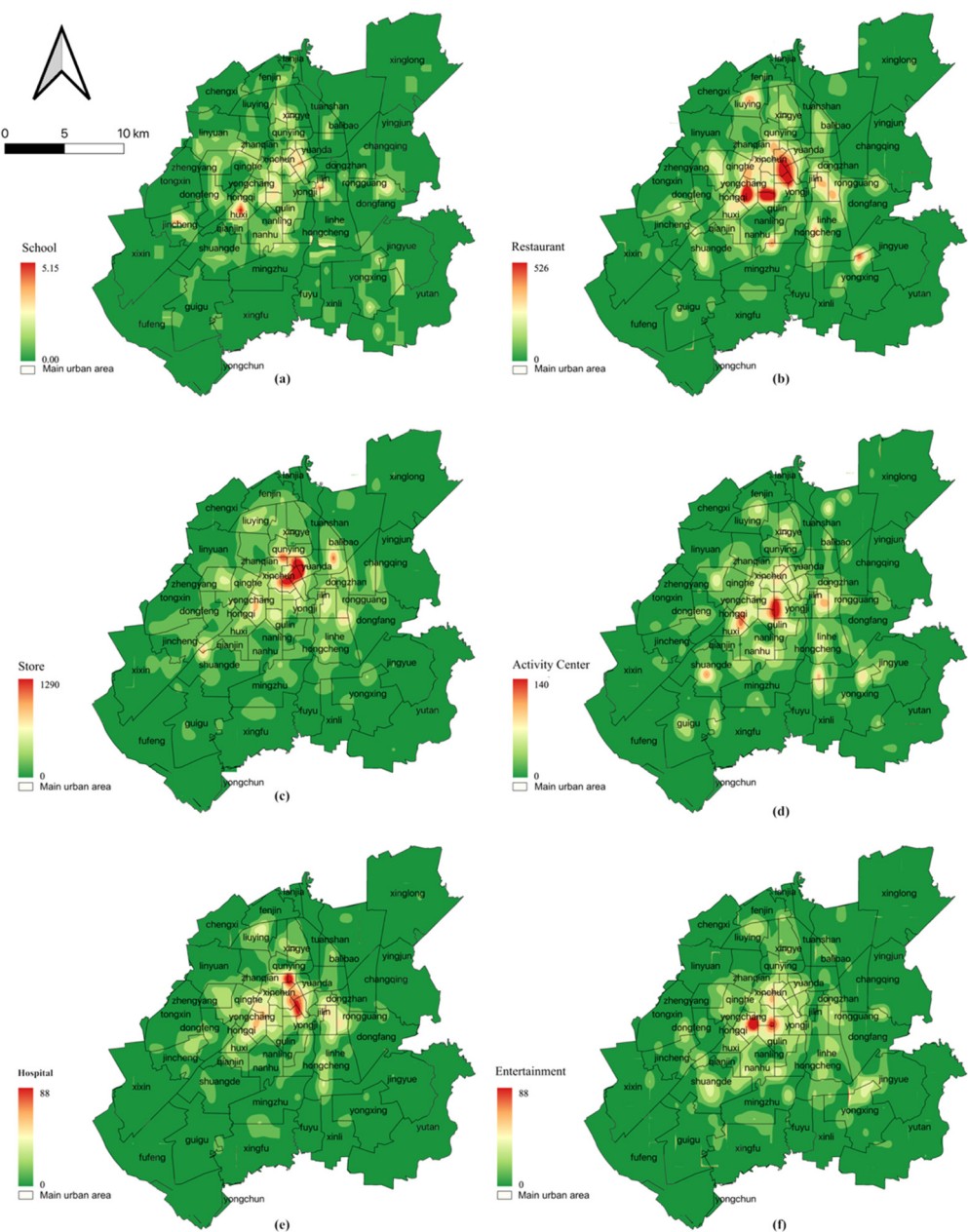

**Figure 9.** Kernel density of various service facilities. (**a**) school; (**b**) restaurant; (**c**) store; (**d**) activity center; (**e**) hospital; (**f**) entertainment.

### 3.2.2. Spatial Distribution of Service Facility Diversity

The diversity of service facilities is an important indicator of regional service capacity. The results indicated that the overall level of service facility diversity of Changchun was high and randomly distributed. The highest value of service facility diversity index in Changchun was found in the periphery of the high kernel density area, which was not consistent with the spatial distribution of the high kernel density area. The correlation between the service facility diversity and the spatial distribution of its quantity was low, which means that the service facility diversity was not necessarily at a higher level in areas with a higher number of service facilities. Figure 8 shows the highest kernel density of service facilities in the "one main and two subs " area, but in Figure 10 the diversity of service facilities is at a medium level. This may be due to the higher land price in the city center area, which has formed a more mature single commercial cluster, and the fact that not all POI data were considered in this paper, which may influence the results. The grids

with lower service facility diversity index were concentrated in the eastern and southern periphery of the main city, which indicated that the service facilities in the eastern and southern areas of the main city were more homogeneous.

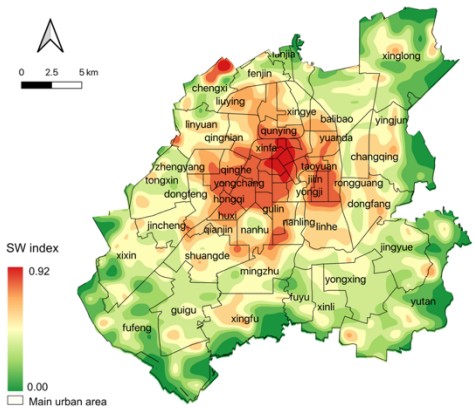

**Figure 10.** Spatial distribution of service facility diversity.

*3.3. Match between Population Activity and Services Facilities*

The first law of geography states that everything is related and that things in close proximity are more closely related [67]. The relationship between things can change with geographical location, which means there is spatial non-smoothness between elements. On the one hand, we investigated the spatial match between population activity intensity and service facility diversity to explore whether there was spatial heterogeneity between the two. On the other hand, based on age characteristics, we analyzed the spatial match between population activity intensity and the number of service facilities for different age groups. In this paper, bivariate spatial correlation analysis was conducted using the GeoDa1.20 software.

3.3.1. Spatial Matching Characteristics

Differences in relationships between population activity intensity and service facility diversity (using the Shannon-Weiner diversity index) were explored using characterizations of quadrants as shown in Figure 11. The horizontal coordinate in Figure 11 is the standardized value of service facility diversity of the grid, the vertical axis is the mean value of population activity intensity normalized for adjacent grids, the origin is the mean value of the horizontal and vertical axes. The red line is the diffusion line of the scattered points, and its slope is the Global Moran's I. Figure 11 indicates that there is a significant positive spatial autocorrelation between population activity intensity and service facility diversity in the main urban area, which means that the higher the activity intensity, the higher the surrounding service facility diversity. Both the service facility diversity and population activity intensity of the scattered points in the first quadrant are higher than the average, indicating that the population activity and service facility diversity in Changchun are spatially positively correlated in a "high-high" manner, i.e., the spatial match between the two is coordinated at a high level. Some of the scattered points fall in the third quadrant, and there is a positive spatial correlation of "low-low" clustering, meaning that the spatial match between the two is coordinated at a low level. The scattered points in the first quadrant and the third quadrant account for 62.20% of the total number of scattered points. Some of the scattered points fall in the fourth quadrant, which shows a negative spatial correlation of "high-low," and there are also some grid points in the second quadrant, which shows a negative spatial correlation of "low-high," These two quadrants indicate that the two spatial matches are not coordinated.

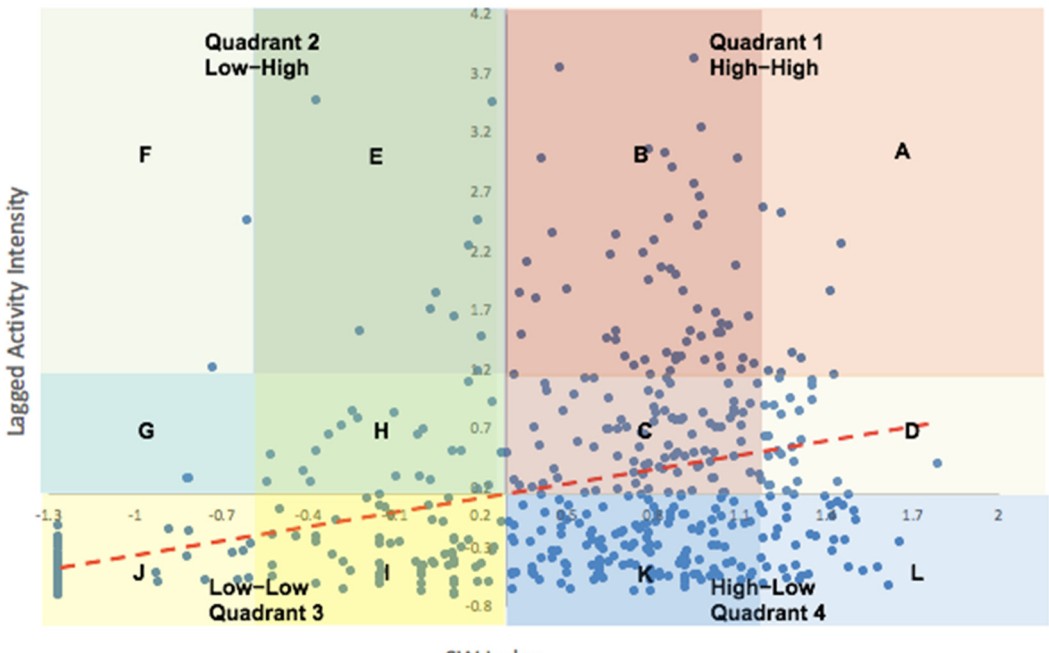

**Figure 11.** Global Moran's I of service facility diversity and population activity intensity. Note: The uppercase letters represent the spatial match type.

The scatter points are distributed in different quadrants, and there are differences in the spatial matching of population activity intensity with the level of service facility diversity. Since the scatter points in quadrant 1 and quadrant 2 in Figure 11 have some outlier points, in order to analyze the spatial differences existing between population activity and service facilities in more detail, we further subdivided the four quadrants into 12 spatial matching types using the mean $\pm$ standard deviation ($\overline{SW} \pm SD$, $\overline{\text{Lagged Activity Intensity}} + SD$). Only $\overline{\text{Lagged Activity Intensity}} + SD$ was used to divide the vertical axis because the data in the third and fourth quadrants were denser.

In the first quadrant, both types A and C have a better overall spatial match, and the service facility diversity and population activity intensity of type A are both higher than that of type C, achieving a higher level of spatial match. The diversity is higher than the intensity of population activity, and the supply is greater than the demand.

The scattered points in the second quadrant mainly fall in type H. The diversity of service facilities in this area is slightly lower than the population activity intensity in the surrounding area, and the service facilities can hardly meet the population demand. Therefore, it is necessary to strengthen the construction of service facility support. A few points fall in type E, where the service facility diversity is much lower than the surrounding population activity intensity, and the degree of spatial incoherence is higher.

In the third quadrant, scattered points in type I represent the spatial coordination between service facility diversity and population activity intensity at a lower level. The scattered points of type J are also at a low level of coordination, but the service facilities are much lower than the intensity of population activity. In addition, there is continuous point clustering on the left side of type J. This is because there are 106 grids in Changchun with a service facility diversity of 0 and a population activity not 0. The service facility diversity is normalized to form an obvious edge boundary with lagged activity intensity scattered points on the left side.

In the fourth quadrant, type K and type L have a large number of scattered points clustered, and the service facility diversity is higher than the population activity intensity. This phenomenon is more obvious in the grid of type L.

According to the Second Law of Geography it is known that spatial segregation causes spatial heterogeneity [68]. Since the global Moran's I index cannot distinguish between

local spatial heterogeneity, we used the local Moran's I index to further explore how the spatial match between service facility diversity and population activity intensity behaves on each grid and to draw LISA maps. The results are shown in Figure 12a,b, where the intensity of population activities and the diversity of service facilities in Changchun show a clear spatial divergence. The characteristics of different types of spatial matching are reported in Table 2.

**Table 2.** Characteristics of the spatial match between service facility diversity and population activity intensity.

| Space Match | Coordination Type | Type | Diversity Index | Activity Intensity | Spatial Distribution | Percentage |
|---|---|---|---|---|---|---|
| Coordination | High-High | A | $SW > \overline{SW} + SD$ | $D > \overline{D} + SD$ | This type of grid is mainly located within the third ring area and is more scattered. | 1.95% |
| | | B | $\overline{SW} < SW < \overline{SW} + SD$ | $D > \overline{D} + SD$ | This type of grid is mainly distributed in the central area with more concentrated distribution. | 10.55% |
| | | C | $\overline{SW} < SW < \overline{SW} + SD$ | $\overline{D} < D < \overline{D} + SD$ | This type of grid is mainly distributed in a circular pattern between the central area and the third ring area. | 7.47% |
| | | D | $SW > \overline{SW} + SD$ | $\overline{D} < D < \overline{D} + SD$ | This type of grid is mainly in Shuangde and Southern New Town, with a more scattered distribution. | 1.95% |
| | Low-Low | I | $\overline{SW} - SD < SW < \overline{SW}$ | $D < \overline{D}$ | This type of grid is mainly distributed in the edge area, with more concentrated distribution in the east and southwest side. | 15.42% |
| | | J | $SW < \overline{SW} - SD$ | $D < \overline{D}$ | This type of grid is mainly distributed in the edge area of the main urban area. | 5.19% |
| Uncoordinated | Low-High | E | $\overline{SW} - SD < SW < \overline{SW}$ | $D > \overline{D} + SD$ | This type of grid is mainly distributed in the central area where the population is concentrated and the distribution is more scattered. | 1.79% |
| | | F | $SW < \overline{SW} - SD$ | $D > \overline{D} + SD$ | This type of grid is mainly distributed in the central area with sporadic distribution. | 0.32% |
| | | G | $SW < \overline{SW} - SD$ | $\overline{D} < D < \overline{D} + SD$ | The local Moran's I of grid points failed the test. | 0.00% |
| | | H | $\overline{SW} - SD < SW < \overline{SW}$ | $\overline{D} < D < \overline{D} + SD$ | This type of grid extends along the Third Ring Road and is more distributed. | 1.95% |
| | High-Low | K | $\overline{SW} < SW < \overline{SW} + SD$ | $D < \overline{D}$ | This type of grid is mainly distributed in the edge area of the main urban area and concentrated in the east. | 8.44% |
| | | L | $SW > \overline{SW} + SD$ | $D < \overline{D}$ | This type of grid is mainly distributed in the edge area of the main urban area. | 1.95% |

Note: D stands for lagged activity intensity. Percentages do not add up to 100% because grids with insignificant local spatial matches are not included in the table for analysis. The table colors correspond to the spatial matching types in Figure 11.

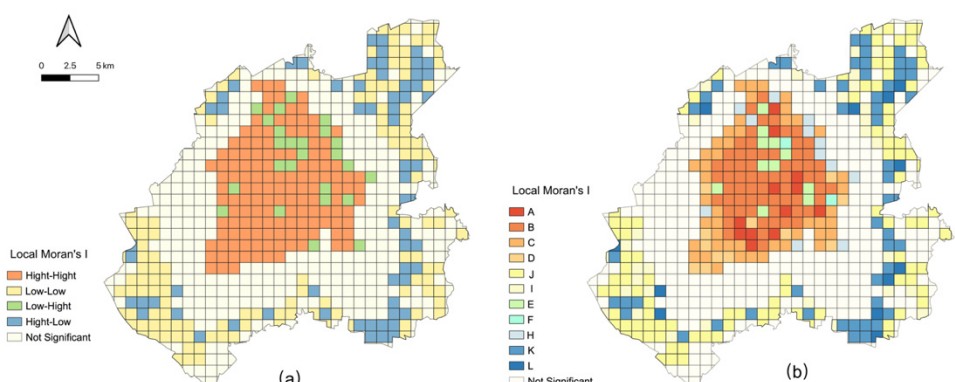

**Figure 12.** Local Moran's I between service facility diversity (**a**) and population activity intensity (**b**).

The "high-high" spatially coordinated grid is concentrated in the Third Ring Area of Changchun, accounting for 21.92% of the total grid. The intensity of activity in this area is higher, and it is the economic and political center of Changchun, so the convenient transportation and perfect service facilities enable people to choose service facilities according to their needs and preferences in a wider range. Further subdividing, we found that type A grids account for about 1.95% and are more scattered. Type B grids are concentrated in the central area, accounting for about 10.55%. The diversity of the service facilities in type B grids can be further improved. Type D grids are mostly distributed in Shuangde and Southern New Town. This area is a key development area planned by Changchun and is an important area for the future population transfer of Changchun.

"Low-low" spatial coordination means that the intensity of population activities and the diversity of service facilities are both at low levels. This type of grid is mainly distributed in the edge of the main city, accounting for 20.62% of the total grid. In order to facilitate commuting and medical care, people tend to choose the central area where they can meet a variety of needs, with convenient transportation and a wide range of services and facilities. Therefore, the farther away from the central area, the lower the intensity of its population activities. At the same time, the location of service facilities will tend to favor densely populated areas with convenient transportation. The edge of the main urban area has lower land prices than the central area and has a good ecological environment, which is more suitable for the distribution of industrial, health, and retirement facilities. To sum up, the population activity intensity and the service facility diversity in the peripheral areas are both at a lower level. Specifically, Type I grids account for the highest proportion of the total grids, with a proportion of 15.42%. Type J grids account for 5.19% of the total grids, which are more scattered and mainly distributed in the peripheral areas.

The "low-high" spatially uncoordinated grids are mainly distributed in the central area and scattered along the Third Ring Road, with a total of 25 grids accounting for 4.06% of the total grids. Specifically, Type E is mainly distributed in the central area where the population is concentrated, with large service facilities such as large hospitals, railway stations, and parks, etc. Although the type of service facilities in the grid is relatively homogeneous, the attractiveness to the population is high and the population activity is higher. The type H grid is mainly composed of large residential areas with a high intensity of population activities, and the diversity of service facilities does not match the intensity of population activities.

The "high-low" spatially uncoordinated grids are mainly located at the edge of the main urban area, with 72 grids accounting for 12.01% of the total grids. Most grids of this type are newly built neighborhoods with completed construction of service facilities and low occupancy rate, resulting in spatial incoherence in the fringe areas. Grids of type K are mainly distributed in the eastern fringe areas of the main city, accounting for 8.44% of the total grids. Grids of type L are scattered in the edge areas, and the population in the central area should be encouraged to spread to the eastern edge areas as soon as possible.

3.3.2. Spatial Matching by age group

There were significant differences in the degree of spatial matching between age groups and service facilities (See Table 3). Specifically, the 18–25 year age group is more attracted to entertainment facilities, with a matching ratio of over 0.520. The 25–35 year age group had the highest spatial matching ratio with hospital service facilities. On the one hand, this reflected the fact that hospitals are generally located near dense employment areas, and on the other hand, it also reflected the fact that hospitals are often located in central areas with dense young populations, resulting in a spatial mismatch. The spatial match between the population and restaurant facilities was lower for all age groups, probably because the take-out industry is more developed in China and the correlation between the dining activities of the population and the location of restaurant facilities is lower. The places providing care services for the elderly in China are generally located inside activity centers, so the 65–75 year age group is more likely to go to activity centers in order to receive day care services. The 65–75 year age group is more likely to be spatially matched to entertainment service facilities because China has invested more in medical services for the elderly, the physical quality of the elderly has improved significantly, and the number of trips made by the elderly for recreational purposes is gradually increasing. The 65–75 year age group has more leisure time and can be seen more frequently in parks and other leisure places in the city.

**Table 3.** Spatial matching by age group.

| Category | 18–25 | 25–35 | 35–45 | 45–55 | 55–65 | 65–75 |
|---|---|---|---|---|---|---|
| School | 0.334 | 0.396 | 0.409 | 0.412 | 0.407 | 0.406 |
| Restaurant | 0.354 | 0.449 | 0.442 | 0.430 | 0.419 | 0.401 |
| Store | 0.397 | 0.558 | 0.577 | 0.579 | 0.594 | 0.595 |
| Activity Center | 0.52 | 0.578 | 0.599 | 0.616 | 0.611 | 0.616 |
| Hospital | 0.456 | 0.623 | 0.647 | 0.655 | 0.668 | 0.676 |
| Entertainment | 0.528 | 0.563 | 0.578 | 0.604 | 0.598 | 0.607 |
| Characteristics | With entertainment as the focus, this group prefers places such as cinemas, gyms, and parks. | Entertainment, stores, and hospitals (Many jobs exist around the hospital facilities in Changchun) | | The 45–55 group has some commonality with 35–45 group activities, and this group is one of the main groups providing intergenerational care, so it has a better spatial match with schools and activity centers than other age groups. | | Hospitals are most closely associated with activities in this age group and the degree of this association deepens with age. |

## 4. Discussion and Future Directions

### 4.1. Discussion

Spatial matching of urban population activities and service facilities is an important element of urban planning. By continuously optimizing the spatial layout of service facilities, it can meet the needs of a population with different characteristics and improve the service level and operation efficiency of the city. In this paper, we used mobile phone signaling data and POI data to analyze the spatial characteristics of population activities and service facilities using bivariate spatial correlation analysis.

Our study found a clear temporal pattern in the intensity of the activity of the existing population, which was similar to the results of existing studies [31]. This pattern is influenced by the daily activities of individuals, especially during working hours. Chapin argued that the daily activities of individuals are composed of habitual behaviors such as going to work, going home, and shopping [69]. We found that the spatial clustering characteristics of population activities in Changchun were different from Shi et al. (2020) [35], and the intensity of population activities in small and medium-sized cities was more significantly influenced by the business district and more concentrated. The

intensity of population activity was strongly influenced by the urban layout; for example, the gravitational center of population activity in our study area was in the west. This is because the western part of the main urban area is an important industrial corridor with many jobs, while the eastern part is a composite ecological corridor in the suburbs, which carries out ecological service functions.

The layout of service facilities varied from city to city [70], and the spatial pattern of service facilities was generally divided into concentric circles, sectors, and multi-core layouts [71]. The formation of this pattern may be influenced by factors such as population, policy, location, and geographical conditions. The distribution of service facility diversity followed the "center-periphery" rule, but the diversity of service facilities was not highly correlated with the number of service facilities. Areas with a high number of service facilities may also form a single industry cluster, which will reduce the diversity of service facilities. Therefore, it was more reasonable to analyze the distribution of service facilities from two dimensions.

The types of spatial matches between service facilities and population activities were not only matching and mismatching, but also comprised rarer types that have not been observed in existing studies. For example, we found that in the "high-high" coordination, type A services were coordinated with population activities at a higher level than type C. More practically relevant were the spatial matches of types B and D. Although these two types achieved "high-high" coordination, the government can further optimize the spatial allocation of these types by regulating the layout of population and service facilities in the area.

*4.2. Future Directions*

There are limitations in our study. On the one hand, owing to the characteristics of mobile phone signaling data, we could not identify the individual characteristics, economic characteristics, and household characteristics of the population, which limited us from further analyzing the population demand. In addition, POI data also cannot characterize the scale of service facilities, and it was difficult for us to set appropriate weights for different POIs.

The directions of our future research are: First, the incorporation of individual information, which cannot be identified, such as individual income, education, and marriage, may allow more targeted analyses to support urban planning policy development. Second, the weights of different types of service facilities can be adjusted according to the information of their passenger flow, scale, and level to analyze the capacity of service facilities more comprehensively and provide policy support for optimizing urban space. Finally, according to the third law of geography, similarity calculation can be performed using variables such as income, education level, road network density, and traffic congestion volume to analyze the geographic composition of areas with a high degree of spatial matching and coordination.

**5. Conclusions**

We used multi-source big data to evaluate the spatial matching of urban services and facilities based on demography, analyze the overall level of spatial matching and spatial differences, and provide strategies to solve spatial mismatch based on the evaluation. Our main findings are as follows.

(1) Existing population activity intensity has obvious temporal regularity. The activity curve of the elderly group (65–75 years) is significantly different from that of other age groups, and the overall activity intensity is lower.

(2) The spatial distribution of population activity intensity shows a "center-periphery" distribution. The activity trajectory of the elderly is characterized by obvious clustering, and the activity space is dominated by the central area.

(3) The spatial distribution of service facilities is "one main and two subs". The spatial distribution of different types of service facilities varies.

(4)    The correlation between service facilities kernel density and service facility diversity is low, and the service facility diversity is not certain to be high in areas with higher service facilities kernel density.

(5)    The population activity intensity and the service facility diversity have a good spatial matching degree, but there is also a spatial difference. The "high-high" spatially coordinated grids are clustered in the central areas, while the "low-low" spatially coordinated grids are mainly distributed in the peripheral areas. We can divide the spatial matching into more types and identify the nuances of spatial matching of population activities and service facilities in more depth, which can better provide the basis for spatial planning.

(6)    There are differences in the degree of spatial matching among different age groups and different service facilities.

In order to solve the problem of spatial mismatch between urban service facilities and population activities, we propose the following: First, decentralize medical service facilities in the core area to reduce the attractiveness of the core area and reduce traffic pressure. Second, encourage enterprises to establish community service centers and day care centers for the elderly by renewing and renovating old facilities in order to reduce the construction cost in the core area and diversify to meet the needs of the elderly. Finally, the government can enrich the diversity of service facilities in the fringe areas to relieve the population pressure in the central area and improve the spatial allocation efficiency of the city.

**Author Contributions:** Conceptualization, Y.C. and Y.H.; methodology, Y.C.; software, L.L.; validation, Y.C. and L.L.; formal analysis, Y.C.; investigation, Y.H.; resources, L.L.; data curation, L.L.; writing—original draft preparation, Y.C.; writing—review and editing, Y.H. and Y.C.; visualization, Y.C.; supervision, Y.H. and Y.C.; project administration, Y.C.; funding acquisition, Y.H. All authors have read and agreed to the published version of the manuscript.

**Funding:** This research was funded by Humanities and Social Science Foundation of Ministry of Education of China, grant number 19YJCZH058.

**Institutional Review Board Statement:** Not applicable.

**Informed Consent Statement:** Not applicable.

**Data Availability Statement:** Not applicable.

**Conflicts of Interest:** The authors declare no conflict of interest.

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
