# Peer review of "Demography-Oriented Urban Spatial Matching of Service Facilities: Case Study of Changchun, China"

_land, doi:10.3390/land11101660_

Round 1

Reviewer 1 Report

1. The spatial matching of population activities and service facilities is a very interesting topic. However, in urban planning, the allocation of service facilities is considered in terms of quantity, layout and timing. What is the difference between this study and this?

 2. In the "Abstract" section, it is suggested to explain the main data and research methods.

 3. In the "Introduction" section, the relevant literature on the influence of service facilities on urban layout and development trend is supplemented; In particular, related literature on the layout of service facilities in the context of smart cities. In addition, the interaction between population activities and service facilities, as well as the contradiction between urban planning standard allocation (1000 persons index).

 4. In the part of "2.1  Study Area", it is better to have map of location and city status, so that readers can quickly understand the research area.

 5.  "2.2 Data"section, the specific source of data should be explained, and the overall representative of data samples should be considered. China currently has three basic telecom operators: China Telecom, China Mobile and China Unicom.

 6. "3. Study results" section, supplementary analysis chart legend.

 7. It is suggested to add a "discussion"section. Based on the existing research results and existing research literature, analyze, discuss and extrapolate 2-3 topics to put forward the basis for the research conclusion.

 8. Research conclusions should be generalized and of general significance on the basis of case studies. Suggestions to deepen.

 9. It is also suggested to supplement the research deficiencies and further research directions.

10. Check text and grammar.

Author Response

Dear Reviewer,

Thanks for your letter and for the reviewers’ comments concerning our manuscript entitled “Spatial Matching of Population Activities and Service Facilities: A Case Study of City” (land-1863340). Those comments and suggestions are helpful for revising and improving our paper. We have carefully made corrections and revised the manuscript which we wish to be approved. Please see the attachment for details of the revision.

Thank you very much for your help.

Kind regards,

Dr. Hu,

E-Mail: [email protected]

Reviewer 2 Report

Comments provided in the attached file.

Author Response

Dear Reviewer,
Thanks for your letter and for the reviewers’ comments concerning our manuscript entitled “Spatial Matching of Population Activities and Service Facilities: A Case Study of City” (land-1863340). Those comments and suggestions are helpful for revising and improving our paper. We have checked and improved the English writing in the revised manuscript. We have carefully made corrections and revised the manuscript which we wish to be approved.
Please see the attachment for details of the revision.
Thank you very much for your help.
Kind regards,
Dr. Hu,
E-Mail: [email protected]

Reviewer 3 Report

The topic of this paper is relevant to the journal. However, several aspects need more details and I would like to share with my specific comments and suggestions below:

1.      The literature review focuses on the review of population activities, but the review of the evaluation methods of service facilities and the relationship with population activities is not sufficient.

2.      Lines 180-184 has been explained in the introduction to the research area, so there is no need to explain it again. Lines 186-188 are also similar to the previous ones. What are the abbreviations of PS and CS?

3.      From the title of the article, spatial matching of population activities and service facilities is the focus of the study, but the author only uses a few words to describe the relationship between the two, and most of them focus on the single phenomenon itself.

4.      The significance of the policy should not be limited to Changchun itself, but its inspiration to other cities at home and abroad is worthy of readers' interest

Author Response

(The authors gave the same response as above.)

Round 2

Reviewer 2 Report

The authors have enthusiastically revised the manuscript and it shows in the fluid presentation, more detailed and richer interpretations, they have been able to bring to their analyses and conclusions. Best wishes to the authors with their future studies and publications. Well done!

Author Response

Dear Reviewer,

Thank you again for your letter and the enthusiastic comments of the reviewers on our manuscript entitled “Demography-Oriented Urban Spatial Matching of Service Facilities: Case Study of Changchun, China” (land-1863340). Those comments are all valuable and very helpful for revising and improving our paper as well as the important guiding significance to our researches. We have studied comments carefully and have made correction which we hope meet with approval.

Please see the coverletter for Reviewer for details of the revision.
Thank you very much for your help.
Kind regards,
Yaqi Hu 
Corresponding author: Yaqi Hu 
E-mail: [email protected] 
